# VISUAL TRANSFORMATION TELLING

## ABSTRACT

Humans can naturally reason from superficial state differences (e.g. ground wetness) to transformations descriptions (e.g. raining) according to their life experience. In this paper, we propose a new visual reasoning task to test this transformation reasoning ability in real-world scenarios, called **V**isual **T**ransformation **T**elling (VTT). Given a series of states (i.e., images), VTT requires to describe the transformation occurring between every two adjacent states. Different from existing visual reasoning tasks that focus on surface state reasoning, the advantage of VTT is that it captures the underlying causes, e.g. actions or events, behind the differences among states. We collect a novel dataset which comprise 13,547 samples to support the study of transformation reasoning. Each sample involves several key state images along with their transformation descriptions. Our dataset spans diverse real-world activities, providing a rich resource for training and evaluation with automated, human, and LLM assessments. To construct an initial benchmark for VTT, we test models including traditional visual storytelling (CST, GLACNet) or dense video captioning methods (Densecap) and advanced multimodal large language models (LLaVA v1.5-7B, Qwen-VL-chat, Gemini-1.5, GPT-4o, and GPT-4), as well as their upgraded versions based on our learning on human reasoning. Experimental results reveal that even state-of-the-art models still have a significant gap with human performance in VTT, highlighting substantial areas for improvement.

## 1 INTRODUCTION

What comes to your mind when you are given a series of images, e.g. Figure 1? We may first notice the content of each image, then connect them in our mind, and finally conclude a series of events from images, i.e., the entire intermediate process of cooking noodles. In fact, as described in Piaget's theory of human cognitive development Bovet (1976); Piaget (1977), this is a typical reasoning process from states (i.e., single images) to transformation (i.e., changes between images). This ability, perceiving and analyzing transformations between states, marks a significant advancement in cognitive development. In the preoperational stage (2-7 years old), children tend to concentrate on static states and often overlook these dynamic transformations. However, as they enter the concrete operational stage (7-12 years old), their cognitive capabilities evolve, enabling them to gradually appreciate and understand the transformations between states.

Interestingly, the development of computer vision, especially at the stage of deep learning, follows a similar pattern. Early computer vision primarily focused on tasks such as image classification, image detection, image captioning, image question answering, and image generation, aiming to understand or generate static states, and it has achieved satisfactory results. Recent multimodal large language models (MLLMs) Liu et al. (2023a); Bai et al. (2023); et al. (2024a;b) have further benefited from larger data volumes and more extensive model parameters, achieving even greater breakthroughs. As machines' ability to understand and generate static states approaches or surpasses human levels, researchers have shifted focus to dynamic vision tasks. These include visual storytelling Ting-Hao et al. (2016), procedure planning Chang et al. (2020), and video generation Singer et al. (2022); Ho et al. (2022); Hong et al. (2022). Despite recent advances, current models often struggle to accurately understand and represent transformations, leading to errors in visual content interpretation and generation. For example, Sora Liu et al. (2024), while capable of producing high-quality videos, faces challenges in modeling basic transformations such as glass breaking. It might display water spilled on the table before the glass itself breaks, indicating a failure to capture the sequential transformation. This limitation highlights the critical need for more robust transformation modeling to tackle complex visual reasoning tasks effectively.

In this paper, we propose a new task, called **V**isual **T**ransformation **T**elling (VTT), to directly evaluate the ability of transformation modeling in real scenarios. VTT task asks models to generate sentences to describe the transformation for a given series of states, i.e. images. Different from traditional visual reasoning tasks that only consider state differences, VTT focuses on digging for underlying transformation behind observation. As the images $s_3, s_4$ shown in Figure 1, the

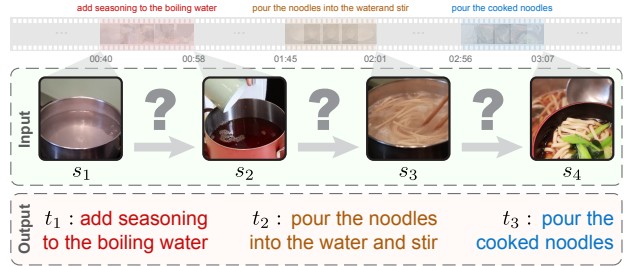

Figure 1: An example of **Visual Transformation Telling.** Given a series of *states (images)*, the goal is to reason and describe *transformations* between every two adjacent states.

change in the position of noodles is merely a surface phenomenon, the more fundamental reason is that someone pouring out the noodles, leading to the state transition. Previously, there have been some preliminary studies Park et al. (2019); Hong et al. (2021); Qiu et al. (2023) on transformation. However, they are defined in an artificial environment with extremely simple transformations, which is difficult to simulate the diversity and complexity of transformations in reality. In contrast, our dataset covers a wide range of daily activities from two extensive instructional video collections, CrossTask Zhukov et al. (2019) and COIN Tang et al. (2019; 2021), which include temporal boundaries and descriptions annotations. These annotations, originally intended for tasks like step localization and action segmentation, were leveraged to structure the data for our Visual Task Transformation task. Specifically, key video frames were extracted to serve as state inputs, while the annotated descriptions of the main steps were employed as transformation targets.

We benchmark existing models on VTT tasks and conduct extensive analysis. Given the similarity between VTT and visual storytelling and dense video captioning, i.e., both of which output a sequence of sentences based on a series of keyframes, we adapt several typical methods, including CST Gonzalez-Rico & Fuentes-Pineda (2018), GLACNet Kim et al. (2019), and Densecap Johnson et al. (2016). Additionally, we evaluate several multimodal large language models (MLLMs), including open source models, i.e., LLaVA v1.5-7B Liu et al. (2023a), Qwen-VL-chat Bai et al. (2023), and close source models, i.e., Gemini-1.5 et al. (2024a), GPT-4 et al. (2024b) and GPT-4o Hel. Experimental results indicate that existing models still have significant scope for improvement. According to the human and LLM evaluation, even the best performing model, i.e., Gemini-1.5, achieves scores of only 3.95, and 4.17 (out of 5) in terms of Relevance, and Logical Soundnes, highlighting a significant gap compared to human performance. We further perform qualitative analyses on test cases, identifying four common error types in MLLMs: bias, misidentification, hallucination, and illogicality. We further explore strategies to improve existing model on VTT data. We find that fine-tuning MLLMs on VTT datasets significantly improves both relevance and logical consistency, suggesting that existing training data lack sufficient information for effective transformation reasoning. Prompt strategies like forcing the model to predict the overall transformation topic can improve the performance and alleviate hallucination problems. Moreover, while explicitly modeling differences between states has demonstrated substantial improvements in traditional models, applying similar approaches to MLLMs remains non-trivial, indicating a potential direction for future study.

The contributions of this study are as follows: 1) We introduce a novel visual transformation telling task and collect a dataset to resolve the limitations of transformation reasoning in real-world scenarios. We support this with a comprehensive evaluation framework, incorporating automated metrics, human assessment, and LLM-based evaluation. 2) We benchmark several models, including traditional models and MLLMs (both open-source and closed-source), revealing significant room for improvement. 3) We identify and categorize common error types in current models, offering insights and potential directions for future research.

## 2 RELATED WORKS

Visual reasoning has been considered as one of the next north star of computer vision Fei-Fei & Krishna (2022), and is constantly being examined by the new multimodal large models that have emerged in recent years. Early visual reasoning tasks mainly focus on state-level reasoning. Spot-the-diff Jhamtani & Berg-Kirkpatrick (2018) represents an initial exploration into the visual differences

between states, highlighting the appearance and disappearance of objects. CLEVR Johnson et al. (2017) and GQA Hudson & Manning (2019) concentrate on object relation and logical reasoning. RAVEN Zhang et al. (2019) and V-PROM Teney et al. (2020) concentrate on the induction and reasoning of graphic patterns. VCR Zellers et al. (2019) and Sherlock Hessel et al. (2022) test the machine's ability to learn commonsense knowledge to answer daily questions. In addition to these tasks, there is a series of works related to dynamic reasoning. Physical reasoning Melnik et al. (2023) evaluates the ability to learn physical rules from data to answer questions or solve puzzles. VisualCOMET Park et al. (2020) requires reasoning beyond the given state to answer what happened before and will happen next. Visual storytelling Park et al. (2020) requires logically telling a story from information-incomplete states. The field of visual reasoning tends to shift from static scenes to dynamic ones. While reasoning in dynamic scenes, state and transformation are both crucial, we focus on transformation reasoning to better evaluate and improve this ability, which distinguishes VTT from state-only and more complex composite tasks.

To the best of our knowledge, there are few studies on designing specific tasks for visual transformation reasoning. TVR Hong et al. (2021) and OVT Qiu et al. (2023) require to predict a sequence of property (e.g. color) changes given the initial and final states. However, the synthetic scenario used in both datasets is far from reality and the property changes are not commonly used to describe transformations in real life. In contrast, VTT emphasizes event-level description, which is a more natural way of describing transformations. Visual storytelling Ting-Hao et al. (2016); Ravi et al. (2021) indeed requires event-level description, but transformations are mixed throughout the story, making it difficult to evaluate transformation reasoning specifically. Visual abductive reasoning Liang et al. (2022) has a similar core idea to VTT, which is to find the most likely explanation for incomplete observations. However, VTT aims to reason multiple logically related transformations from states, while their task only requires reasoning a single missing transformation from multiple transformations. Procedure planning Chang et al. (2020) aims to complete a job given states, while VTT focuses on explaining transformations between states, which has wider scenarios, such as explaining the wet ground with rain. Furthermore, the requirement for natural language generation in VTT leads to different evaluations and unique challenges, such as generalization on language compositions and transformation combinations. Finally, walkthrough planning Chang et al. (2020) has a different target, which is to predict intermediate states.

Another topic related to VTT is visual description. Tasks that describe a single image include image captioning Farhadi et al. (2010); Kulkarni et al. (2011), dense image captioning Johnson et al. (2016), and image paragraphing Krause et al. (2017), which vary in the level of detail required. Tasks that describe videos include video description Venugopalan et al. (2015), video paragraph description Yu et al. (2016), grounded video description Zhou et al. (2019), dense video captioning Krishna et al. (2017), and video timeline modeling Liu et al. (2023b) start to describe events rather than a single state. For example, dense video captioning asks to predict temporal boundaries of key events and describe them. However, these tasks do not explicitly require reasoning about transformations since they provide the full process of transformation throughout frames.

## 3 VISUAL TRANSFORMATION TELLING DATASET

### 3.1 TASK DEFINITION

Visual transformation telling aims to test machines' ability to reason and describe transformations from a sequence of visual states, i.e., images. Formally, $N + 1$ images $S = \{s_n\}_{n=1}^{N+1}$ are provided, which are *logically related* and *semantically distinct*. Logically related means these images are associated with a particular event and are arranged in time sequence. Semantically different means that adjacent images come from two discontinuous time points and the content they contain has substantially changed, i.e., a transformation. The objective is then to reason $N$ transformations $T = \{t_n\}_{n=1}^{N}$ between every two adjacent images and describe them in natural language, such that $s_1 \to t_1 \to s_2 \to \cdots \to t_n \to s_{n+1}$ is logically sound.

### 3.2 VTT DATASET CONSTRUCTION

**Data collection.** To create a comprehensive dataset of real-world transformations, we chose instructional videos due to their detailed depiction of everyday activities. Specifically, we used two

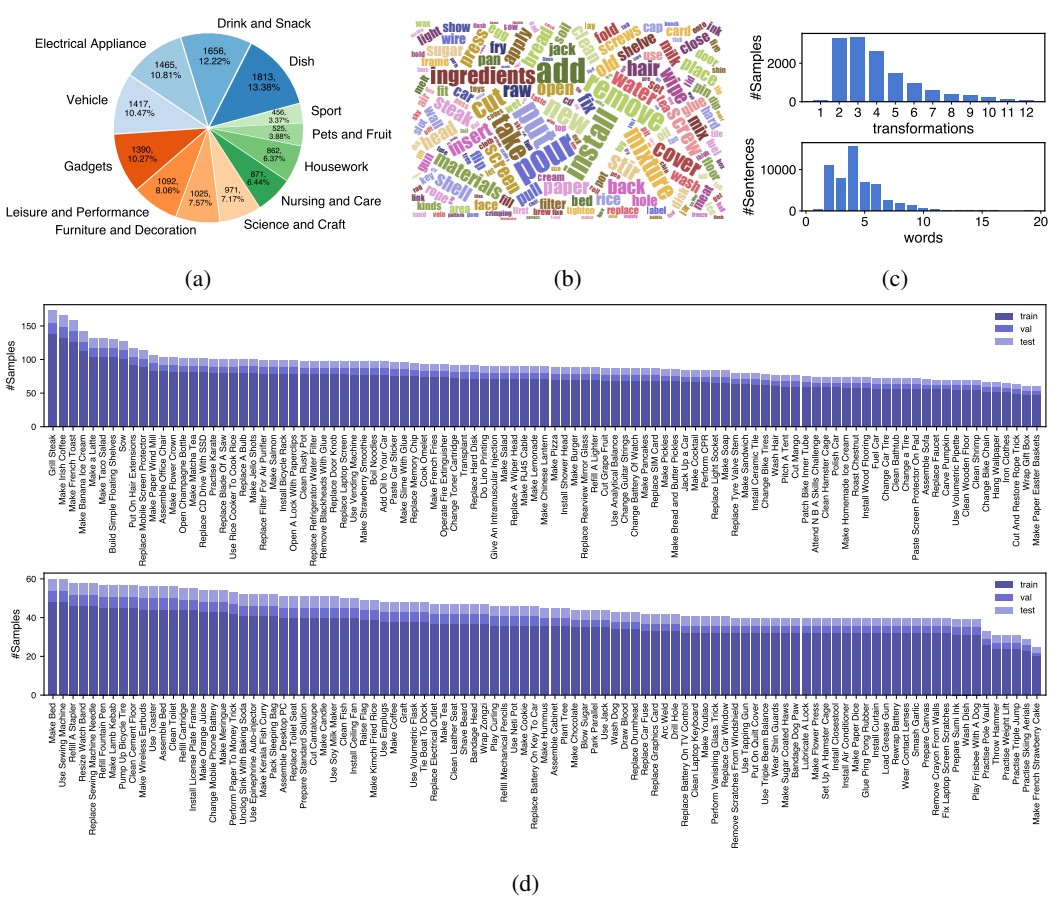

(a)  (b)  (c)

(d)

Figure 2: Distributions of VTT samples. (a) Category. (b) Words. (c) Transformation length (top), sentence length (bottom). (d) Topic.

well-known public instructional video datasets: CrossTask Zhukov et al. (2019) and COIN Tang et al. (2019; 2021). These datasets provided a rich source of data for our VTT dataset.

**State and transformation description.** Figure 1 illustrates an instructional video from COIN on cooking noodles and how we transformed their annotation into our VTT dataset. We can see that the video is segmented into multiple main steps, each annotated with precise temporal boundaries and text labels. For **state image extraction**, the best choice is the frame just before or after a transformation. CrossTask's and COIN's precise temporal segment annotations, which undergo three rounds of refinement Tang et al. (2019), can satisfy this requirement. For the first transformation, we used the first frame of the corresponding step segment as its start state and the last frame as its end state. For the remaining transformations, the end state is extracted in the same way, while the start state shares the end state of the previous transformation. We filter out samples containing too similar adjacent states based on CLIP [1] feature similarity to avoid situations where the transformation cannot be recognized. We also use EasyOCR [2] to filter out samples containing characters in the image to avoid potential caption leakage. For **transformation descriptions**, we used original text labels as transformation labels. We manually checked the quality of 200 random samples and found that transformations could be reasoned out from states most of the time. Using this method, we collected 13,547 samples with 55,482 transformation descriptions from CrossTask and COIN, forming our new data for VTT.

**Category and topic labels.** The VTT dataset also includes annotations such as *category*, *topic*, and *transformation description*, which are collected and organized from CrossTask and COIN. Step

---

[1] https://huggingface.co/openai/clip-vit-large-patch14
[2] https://github.com/JaidedAI/EasyOCR

labels and corresponding segments are provided by both datasets. In CrossTask, step labels were derived from WikiHow, whereas COIN employed experts to define them. Annotators were then tasked with labeling the step categories and corresponding segments for each video. We collected and organized these annotations in a uniform format for the VTT dataset. Both CrossTask and COIN provide topic information, which pertains to the task to be solved. COIN also provides categories as domain information, which are absent in CrossTask. We manually classify all topics from CrossTask into existing categories. Table 6 in Appendix shows the full list of 12 categories and 198 topics.

**Dataset Split and Statistics.** We randomly split the data into Train/Val/Test sets with 10,759, 1,352, and 1,436 samples at the topic level. The detailed topic distribution is shown in Figure 2d, indicating that about half of the topics have over 100 samples. The main statistics of the VTT dataset are summarized in Table 1. VTT also requires models to generalize to handle transformation

Table 1: VTT dataset statistics.

|  | CrossTask | COIN | Train | Val | Test | Total |
|---|---|---|---|---|---|---|
| Categories | 4 | 12 | 12 | 12 | 12 | 12 |
| Topics | 18 | 180 | 198 | 198 | 198 | 198 |
| Samples | 1825 | 11722 | 10759 | 1352 | 1436 | 13547 |
| States | 12860 | 56169 | 54716 | 6974 | 7339 | 69029 |
| Trans. | 11035 | 44447 | 43957 | 5622 | 5903 | 55482 |
| Unique Trans. | 105 | 749 | 853 | 812 | 806 | 853 |

combinations not present in the training set. Figure 2 illustrates the distribution of the sample categories, keywords, transformation length, and sentence length of VTT. The category distribution and word cloud reveal that VTT encompasses a wide range of daily activities. The distribution of transformation length shows diversity and most samples involve 2-5 transformations. The average sentence length is around 2-6 words, suggesting that brief descriptions are predominant.

## 4 BENCHMARK ON VTT

### 4.1 MODEL SELECTION

**Traditional models.** We first adapt two classic visual story telling methods for comparison, including CST Gonzalez-Rico & Fuentes-Pineda (2018) and GLACNet Kim et al. (2019), which are both winners of the visual storytelling challenge Mitchell et al. (2018). This is because visual storytelling generates $N$ descriptions from $N$ images, that is similar to our VTT task. In addition, we also compared with a dense video captioning method called DenseCap Johnson et al. (2016), since dense video captioning also has a similar visual description target, which aims to describe a series of events in a video and requires predicting temporal boundaries for events. All methods were closely implemented as per the original paper. For a better image understanding, we also provided baseline models with CLIP as image encoder marked with '*'. The implementation details of TTNet as well as the baseline models are described in the supplementary.

**Multimodal language models.** MLLMs have shown promising capabilities on various vision language benchmarks. To test how well they perform on VTT, we test two open-source models, including LLaVA v1.5-7B Liu et al. (2023a), Qwen-VL-chat Bai et al. (2023). We also test four closed source models through their public API, including Gemini-1.5 et al. (2024a), GPT-4 et al. (2024b), and GPT-4o Hel. Considering that these models may not be well adapted to the task form of VTT, such as language style, differences in word usage, etc., we also tune the LLaVA model with LORA Hu et al. (2021) on VTT for testing.

### 4.2 EVALUATION PROTOCOL

**Automated metrics.** We follow previous works on visual descriptions Ting-Hao et al. (2016); Krishna et al. (2017); Liang et al. (2022), and select common used metrics for evaluation, including BLEU@4 Papineni et al. (2002), CIDEr Vedantam et al. (2015), METEOR Banerjee & Lavie (2005), ROUGE-L Lin & Hovy (2002), SPICE Anderson et al. (2016), and BERT-Score Zhang et al. (2020),

**Human evaluation.** For automatic evaluation metrics, factors such as vocabulary choice, sentence structure, and sentence length can impact scores, even for semantically identical sentences. As this is the first introduction of this benchmark, we prioritized accuracy through human evaluation. We asked 25 human annotators to assess the quality of transformation descriptions using a Likert scale ranging from 1 to 5 based on the following criteria: *fluency*, measuring the clarity and coherence of

Table 2: Results on VTT evaluated using B@4(BLEU@4), M(METEOR), R(ROUGE-L), C(CIDEr), S(SPICE), BS(BERT-Score), Flu.(Fluency), Rel.(Relevance), and Logic.(Logical Soundness). * indicates using CLIP as image encoder. 'Sep' and 'multiturn' means inputting each image in one prompt separately and providing each adjacent pair in multiple prompt step-by-step.

| Model | B@4 | M | R | C | S | BS | Flu. | Rel. | Logic. |
|---|---|---|---|---|---|---|---|---|---|
| Human | 11.79 | 13.66 | 29.49 | 82.26 | 24.41 | 40.95 | 5.00 | 4.88 | 4.88 |
| CST | 10.09 | 11.39 | 25.98 | 43.22 | 9.28 | 16.30 | - | - | - |
| CST* | 13.96 | 19.21 | 38.11 | 84.60 | 21.85 | 25.66 | 2.04 | 3.16 | 2.96 |
| GLACNet | 42.77 | 45.26 | 52.98 | 381.48 | 45.33 | 60.12 | - | - | - |
| GLACNet* | 55.24 | 59.48 | 66.25 | 508.18 | 60.21 | 71.13 | 4.75 | 3.82 | 3.78 |
| DenseCap* | 48.25 | 52.00 | 59.79 | 439.68 | 53.73 | 66.30 | 4.74 | 3.67 | 3.59 |
| GPT-4 | 4.73 | 6.74 | 11.76 | 28.24 | 11.66 | 25.84 | - | - | - |
| GPT-4o | 4.84 | 6.91 | 12.03 | 29.69 | 13.01 | 28.38 | - | - | - |
| Gemini-1.0 | 8.36 | 10.25 | 19.82 | 47.79 | 16.13 | 31.43 | - | - | - |
| Gemini-1.5 | 8.51 | 11.1 | 20.62 | 52.25 | 17.93 | 33.88 | 4.95 | 3.95 | **4.17** |
| Gemini-1.5 (multiturn) | 8.20 | 9.91 | 19.87 | 42.69 | 16.47 | 31.08 | - | - | - |
| Qwen-VL-chat | 4.71 | 4.57 | 10.62 | 15.32 | 6.25 | 23.93 | - | - | - |
| Qwen-VL-chat (Sep) | 4.70 | 5.62 | 11.23 | 21.91 | 9.38 | 25.64 | - | - | - |
| LLaVA-1.5-7B | 3.06 | 3.30 | 7.19 | 12.04 | 5.18 | 23.21 | - | - | - |
| LLaVA-1.5-7B+Topic | 3.14 | 3.46 | 7.56 | 12.49 | 5.95 | 23.76 | 4.79 | 2.08 | 3.07 |
| LLaVA-1.5-7B$_{LORA}$ | 31.43 | 32.37 | 40.38 | 268.59 | 33.17 | 49.08 | - | - | - |
| LLaVA-1.5-7B$_{LORA}$+Topic | 33.58 | 34.25 | 41.93 | 289.14 | 35.29 | 50.46 | **4.98** | 3.10 | 3.76 |
| TTNet$_{Base}$ | 55.68 | 60.47 | 67.05 | 515.12 | 61.45 | 72.22 | 4.79 | 4.04 | 3.95 |
| TTNet | **61.22** | **66.31** | **71.84** | **570.63** | **66.20** | **76.25** | 4.78 | **4.10** | 4.11 |

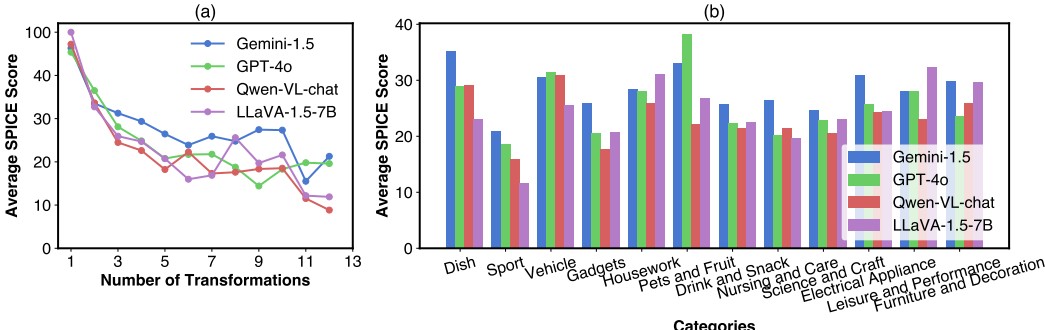

Figure 3: Performance of models under different data: (a) The SPICE values with respect to the number of transformation items. (b) The SPICE values with respect to different categories of data.

the transformations; *relevance*, assessing how relevant the transformations are to the image states; and *logical soundness*, evaluating how well the overall logic aligns with commonsense.

## 5 EXPERIMENTAL RESULTS AND ANALYSIS

In this section, we first summarize the ability of various models on VTT and analyze the performance of MLLMs on different data types. Then, we analyze the error types made by the most advanced MLLMs. Finally, we improve the existing model to preliminarily explore how to model visual transformations better, hoping to inspire future study.

### 5.1 COMPARISON OF BASELINE MODELS

Table 2 summarizes the results of models on the VTT dataset. The results show that both traditional models and SOTA MLLMs have much room for improvement.

For traditional models, GLACNet performs best, which chieves 4.75, 3.82 and 3.78 (out of 5) on Fluency, Relevance and Logical Soundnes respectively. This may because GLACNet uses contextual information more completely.

Among the MLLMs, Gemini-1.5 performs best, achieving scores of 4.95, 3.95, and 4.17 for Fluency, Relevance, and Logical Soundness, respectively. This may be attributed to two factors: First, Gemini employs multimodal interleaving to pre-train from scratch, which contrasts with other MLLMs that primarily rely on knowledge embedded in language models. This direct multimodal pre-training approach may enable Gemini to acquire a more comprehensive knowledge. Second, Gemini's training data includes videos, allowing it to encounter data more similar to VTT scenarios during training. However, it does not demonstrate a substantial advantage over traditional models and still exhibits a significant gap compared to human performance, as indicated by both automated and human evaluations. Since VTT requires understanding across multiple images, we also explored a step-by-step prompting strategy, wherein the model is provided with each adjacent image pair sequentially and asked to describe each transformation. Nevertheless, this multi-turn approach did not yield improved results, potentially due to the increased dependence on historical dialogues, thereby introducing additional complexity.

Further analysis based on human evaluation shows that the main problem with the current large model is inconsistency with the input image, that is, they always generate text that is not completely related or even completely unrelated to the image. In addition, the output of MLLMs also have logical errors, which are manifested in the generated activities violating commonsense or the generated transformations sequence is unreasonable. Even tuning cannot solve these problems well, indicating that more efforts are needed.

## 5.2 Performance across Different Data Types

We further analyze the model's performance across different data types. As shown in Figure 3, for all MLLMs, an increase in the number of transformations correlates with a decline in performance, indicating that the models struggle to manage long contexts effectively. This drop in performance may be due to the models' difficulty in modeling long-range dependencies, as the complexity of reasoning increases with the number of transformations. Longer sequences require maintaining coherence and tracking intricate changes over multiple steps, which current MLLMs may not handle efficiently due to limitations in their attention mechanisms or insufficient training on extended contextual data.

In examining performance across event categories, we observe that the specific types in which different models excel are inconsistent, likely due to variations in the training data distribution. However, one consistent finding across all models is that their performance is weakest in the sports category. This suggests that incorporating more relevant data may be necessary to enhance model performance for this particular type.

## 5.3 Qualitative Analysis and Common Error Types

We qualitatively analyze the output of different MLLMs and show some examples in Figure 4 (more cases can be found at Appendix). We summarize the common errors into four types:

**Bias:** Models can be misled by the presence of specific objects to conclude that certain non-occurring events are happened. As the example of the event 'cut mango', the simultaneous appearance of the glass and the fruit leads the Qwen and LLaVa to assume that the event is related to juicing. This type of error indicates that the models are overly reliant on co-occurrence patterns observed in the training data, which may not accurately reflect real-world scenarios.

**Misidentification:** Models sometimes mistakenly identify objects in images. For instance, LLava failed to recognize contact lenses and incorrectly identified cleaner as lotion. Such recognition errors are more prevalent in models with smaller parameters. This suggests that model capacity and the training data quality significantly impact the object recognition capability, highlighting the necessity for both larger models and more diverse and comprehensive datasets.

**Hallucination:** Models sometimes generate predictions that deviate from the image context, despite they correctly identify objects and topics. This results in the generation that is relevant to the topic but inconsistent with the image, or even generating objects that do not exist. As the example of the

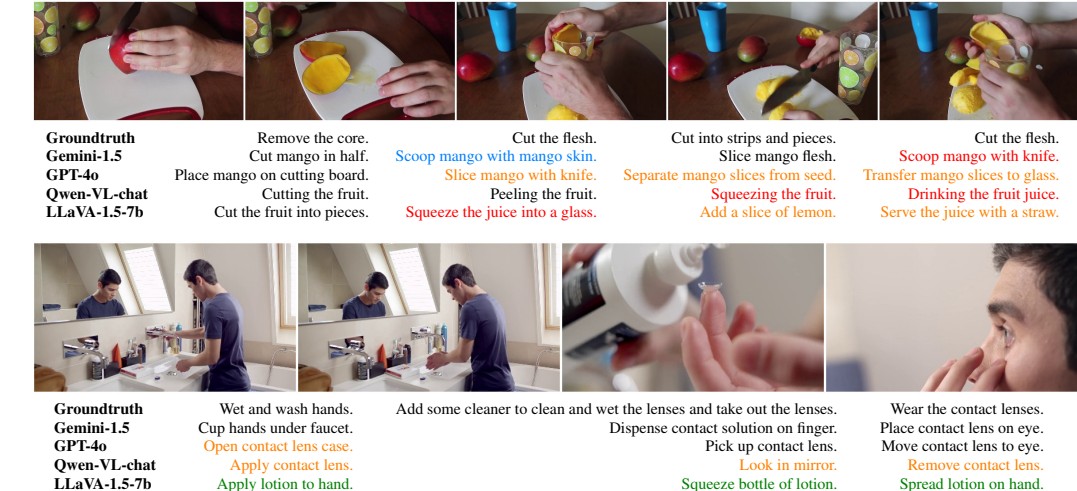

Figure 4: Qualitative comparison on the VTT test data. Above: cut mango. Below: wear contact lenses. Different error types are marked with different colors: bias (red), misidentification (green), hallucination (orange), and illogicality (blue).

event 'wear contact lenses', the output of GPT-4o is consistent with the topic but includes 'contact lens case', which is not present in the image. This issue points to a disconnect between the language and vision components of current MLLMs.

**Illogicality:** Models may output illogical content or even violate commonsense. For example, Gemini outputs 'scoop mangoes with mango skin', which is an implausible scenario. These errors highlight the limitations of models in understanding and applying commonsense reasoning, indicating a need for incorporating more advanced reasoning capabilities and better grounding in real-world knowledge.

## 5.4 FURTHER EXPLORATION

Building on our understanding of the basic pipeline for human reasoning about transformations from visual states, we explore ways to enhance models' capacity for visual transformation reasoning. Given the need for both flexibility and manageable computational overhead, we focus on improving the best-performing traditional model, GLACNet. To further enhance image understanding, we replace the original image encoder with CLIP Radford et al. (2021). We call this improved model TTNet$_{Base}$.

We investigated three key areas for improving the model: (1) **Difference Sensitive Encoding (Diff.):** In addition to the original representation of each state, we include the differences between every pair of adjacent states at the embedding level to enhance the model's ability to capture semantic-level differences between states. (2) **Masked Transformation Modeling (MTM):** To enable the model to fully utilize information from all states and transformations across different steps, we employ a masked transformation modeling strategy. (3) **Auxiliary Learning (Aux.):** we introduce topic prediction and category prediction task for each state series to reinforce the consistency of model outputs with the overall themes. We refer to this improved model as TTNet. Further details can be found in the Appendix D.

The overall performance of TTNet on the VTT task is presented in the last two rows of Table 2, while the ablation study results for each component are shown in Table 3. The results indicate that using the state feature difference provides the most substantial improvement, suggesting that capturing differences is essential for effective transformation reasoning. The subsequent four rows show the results of various combinations of these strategies, and it is evident that utilizing all three strategies yields the best performance. We also evaluate the impact of different auxiliary tasks. From Table 4, topic classification proves more effective than category classification, likely because topics offer a more fine-grained level of information than categories. Notably, using both classification tasks concurrently enhances overall performance.

We also try to apply improved strategies to LLaVA. Considering both 'difference sensitive encoding' and 'masked transformation modeling' require fine-tuning the model to adapt to inputs not

Table 3: Results of applying different key components of TTNet. The first row presents the base model's performance.

| Diff. | MTM | Aux. | B@4 | M | R | C | BS |
|---|---|---|---|---|---|---|---|
| | | | 55.68 | 60.47 | 67.05 | 515.12 | 72.22 |
| √ | | | 59.89 | 64.61 | 70.30 | 556.85 | 75.00 |
| | √ | | 56.26 | 60.92 | 67.57 | 520.04 | 72.72 |
| | | √ | 56.37 | 61.18 | 67.85 | 521.93 | 72.97 |
| √ | √ | | 60.39 | 65.38 | 70.99 | 562.25 | 75.62 |
| √ | | √ | 60.38 | 65.50 | 71.14 | 562.83 | 75.72 |
| | √ | √ | 56.91 | 61.89 | 68.45 | 527.62 | 73.54 |
| √ | √ | √ | **61.22** | **66.31** | **71.84** | **570.63** | **76.25** |

Table 4: Ablation study results on the auxiliary tasks, i.e., category prediction, and topic prediction.

| category | topic | B@4 | M | R | C | BS |
|---|---|---|---|---|---|---|
| | | 60.39 | 65.38 | 70.99 | 562.25 | 75.62 |
| √ | | 59.11 | 64.08 | 69.99 | 549.44 | 74.81 |
| | √ | 60.49 | 65.51 | 71.25 | 562.96 | 75.89 |
| √ | √ | **61.22** | **66.31** | **71.84** | **570.63** | **76.25** |

Table 5: Results of human and LLM evaluations of logical consistency on different models.

| Evaluation | CST | GLACNet | DenseCap | Gemini | LLaVA | LLaVA$_{LORA}$ | TTNet$_{Base}$ | TTNet |
|---|---|---|---|---|---|---|---|---|
| Human | 2.96 | 3.78 | 3.59 | 4.17 | 3.07 | 3.76 | 3.95 | 4.11 |
| Gemini-1.5 | 1.04 | 2.85 | 2.6 | 4.0 | 3.26 | 3.72 | 3.73 | 3.76 |

encountered during pretraining, we opted to implement only 'auxiliary learning' by predicting the corresponding topic. As shown in Table 2, auxiliary learning enhances performance in both the zero-shot and fine-tuned settings. Experiments on traditional models demonstrate that explicitly modeling the differences between states leads to substantial improvement. However, applying similar modeling to MLLMs is not trivial. We leave these improvements for MLLMs to future work.

### 5.5 USING LLM EVALUATION REPLACE HUMAN EVALUATION

For evaluating various aspects, particularly logical consistency, human evaluation remains the most reliable method, as no current metric can precisely measure logical coherence. However, human evaluation is costly and not feasible for large-scale assessments. To address this, we leverage an advanced LLM, Gemini-1.5, to partially substitute for human evaluations by scoring candidate responses. The prompt used for this evaluation can be found in Appendix G. As shown in Table 7, Gemini-1.5 achieves a Spearman's correlation of 88.1 with a p-value of 0.004 when compared to human ratings, indicating a statistically significant correlation. This result suggests that LLMs can serve as a viable proxy for human evaluation to a certain extent.

## 6 CONCLUSION AND DISCUSSION

This paper introduces Visual Transformation Telling (VTT), a novel visual reasoning task that focuses on understanding transformations between states in a series of images, which is a crucial cognitive skill for humans. To the best of our knowledge, this is the first real-world application of transformation reasoning that defines transformation descriptions as outputs. We constructed the VTT dataset, consisting of 13,547 samples, to facilitate this study. We extensively test the capabilities of existing models, both traditional models and state-of-the-art MLLMs. Our experimental results reveal that even the most advanced MLLMs struggle to effectively address this task. We categorize the primary errors of current models into four types: bias, misidentification, hallucination and illogicality. Furthermore, we conduct extensive experiments by tuning MLLMs on VTT data, prompting to force topic generation, and proposing several enhancement strategies for traditional models. Based on our findings, we believe that collecting more data containing explicit transformation information and adapting MLLMs to better understand differences between states (images) represent the most promising future directions for research in transformation reasoning.

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

# A  DATASET SCALE DISCUSSION

As mentioned in the main paper, the limited size of the VTT dataset hinders the generalization ability of current models. Additionally, the dataset covers only a narrow range of transformations, which limits the models' applicability. However, collecting a larger dataset is costly due to the expense of annotating steps/transformations with descriptions and temporal boundaries are expensive. One possible way to mitigate this cost is to use pretrained step localization models Wang et al. (2021); Zhang et al. (2022) or action and object state-recognition models Soucek et al. (2022) to propose coarse steps/transformations and refine the results with human annotators. In addition, we suggest using object state-recognition Soucek et al. (2022) to refine the boundary precision of existing step segments in CrossTask and COIN for constructing larger datasets in the future. Apart from annotating a large-scale dataset, another way is to design a method that can directly learn transformation reasoning from massive raw video-caption data such as HowTo100M Miech et al. (2019). There have already been pioneer works that obtain impressive results on natural language processing tasks, such as GPT-3 Brown et al. (2020) and chatGPT [3], and computer tasks, such as CLIP Radford et al. (2021).

| Category | Topics |
| --- | --- |
| Nursing and Care (14) | Wash Dog, Use Earplugs, Use Neti Pot, Put On Hair Extensions, Use Epinephrine Auto-injector, Perform CPR, Wear Contact Lenses, Remove Blackheads With Glue, Give An Intramuscular Injection, Shave Beard, Wash Hair, Bandage Dog Paw, Draw Blood, Bandage Head |
| Pets and Fruit (7) | Plant Tree, Transplant, Graft, Cut Grape Fruit, Cut Mango, Cut Cantaloupe, Sow |
| Furniture and Decoration (15) | Install Shower Head, Install Ceramic Tile, Install Air Conditioner, Install Curtain, Lubricate A Lock, Replace Door Knob, Install Wood Flooring, Install Closestool, Assemble Cabinet, Assemble Sofa, Replace Faucet, Replace Toilet Seat, Assemble Bed, Build Simple Floating Shelves*, Assemble Office Chair |
| Leisure and Performance (17) | Make Paper Wind Mill, Perform Vanishing Glass Trick, Raise Flag, Play Frisbee With A Dog, Make Chinese Lantern, Carve Pumpkin, Change Guitar Strings, Perform Paper To Money Trick, Pitch A Tent, Open Champagne Bottle, Blow Sugar, Make Paper Easter Baskets, Cut And Restore Rope Trick, Do Lino Printing, Replace Drumhead, Prepare Sumi Ink, Prepare Canvas |
| Dish (23) | Make Kimchi Fried Rice*, Cook Omelet, Make Sandwich, Grill Steak*, Clean Fish, Use Toaster, Clean Shrimp, Make Burger, Make French Toast*, Wrap Zongzi, Make French Strawberry Cake*, Make Pickles, Boil Noodles, Make Bread and Butter Pickles*, Make Kerala Fish Curry*, Make Lamb Kebab, Make French Fries, Use Rice Cooker To Cook Rice, Make Pizza, Make Youtiao, Make Salmon, Smash Garlic, Make Pancakes* |
| Electrical Appliance (20) | Replace Graphics Card, Replace Light Socket, Replace Electrical Outlet, Replace Memory Chip, Use Soy Milk Maker, Change Toner Cartridge, Replace Laptop Screen, Replace Refrigerator Water Filter, Use Vending Machine, Replace Filter For Air Purifier, Replace Hard Disk, Replace Blade Of A Saw, Refill Cartridge, Clean Laptop Keyboard, Arc Weld, Install Ceiling Fan, Replace A Bulb, Paste Screen Protector On Pad, Assemble Desktop PC, Use Sewing Machine |
| Science and Craft (15) | Prepare Standard Solution, Make Flower Press, Use Volumetric Pipette, Hang Wallpaper, Make Candle, Make Soap, Use Triple Beam Balance, Make Flower Crown, Use Volumetric Flask, Paste Car Sticker, Make Slime With Glue, Make Paper Dice, Wrap Gift Box, Set Up A Hamster Cage, Use Analytical Balance |
| Drink and Snack (20) | Make Meringue*, Make Salad, Make Lemonade*, Make Taco Salad*, Make Tea, Make Chocolate, Make a Latte*, Make Homemade Ice Cream, Make Jello Shots*, Make Coffee, Make Cocktail, Make Cookie, Make Irish Coffee*, Roast Chestnut, Make Banana Ice Cream*, Make Orange Juice, Make Matcha Tea, Make Sugar Coated Haws, Make Strawberry Smoothie, Make Hummus |
| Vehicle (21) | Change Bike Chain, Replace Car Fuse, Replace Rearview Mirror Glass, Tie Boat To Dock, Pump Up Bicycle Tire, Change Car Tire, Use Jack, Remove Scratches From Windshield, Jack Up a Car*, Change Bike Tires, Install License Plate Frame, Fuel Car, Replace A Wiper Head, Install Bicycle Rack, Replace Tyre Valve Stem, Change a Tire*, Patch Bike Inner Tube, Polish Car, Replace Car Window, Add Oil to Your Car*, Park Parallel |
| Housework (15) | Put On Quilt Cover, Clean Bathtub, Wash Dish, Clean Leather Seat, Pack Sleeping Bag, Clean Wooden Floor, Clean Toilet, Iron Clothes, Drill Hole, Remove Crayon From Walls, Clean Hamster Cage, Make Bed, Unclog Sink With Baking Soda, Clean Rusty Pot, Clean Cement Floor |
| Sport (10) | Practise Karate, Wear Shin Guards, Practise Triple Jump, Throw Hammer, Play Curling, Practise Skiing Aerials, Practise Pole Vault, Attend N B A Skills Challenge, Glue Ping Pong Rubber, Practise Weight Lift |
| Gadgets (21) | Open A Lock With Paperclips, Replace Mobile Screen Protector, Load Grease Gun, Change Mobile Phone Battery, Replace Sewing Machine Needle, Change Battery Of Watch, Replace SIM Card, Resize Watch Band, Replace CD Drive With SSD, Refill Mechanical Pencils, Make Wireless Earbuds, Refill Fountain Pen, Refill A Lighter, Rewrap Battery, Replace Battery On Key To Car, Fix Laptop Screen Scratches, Operate Fire Extinguisher, Replace Battery On TV Control, Use Tapping Gun, Refill A Stapler, Make RJ45 Cable |

Table 6: The Categories and topics in VTT dataset. Topics marked with * are from CrossTask and others belong to COIN.

---

[3] https://chat.openai.com/

| Metric | Score | Criteria |
|---|---|---|
| Fluency | 5 | All sentences are fluent. |
| | 4 | Most sentences are fluent, with only a few flaws. |
| | 3 | About half of the sentences are fluent. |
| | 2 | Most of the sentences are difficult to read, with only a few being okay. |
| | 1 | All sentences are hard to read. |
| Relevance | 5 | The descriptions are all related to the corresponding before and after images. |
| | 4 | A few descriptions are slightly irrelevant, e.g. the description is related to the underlying topic but cannot be clearly inferred from the images. |
| | 3 | Many descriptions are slightly irrelevant or a few descriptions are irrelevant, e.g. the action or target object mentioned in the transformation does not match the images. |
| | 2 | Many descriptions are irrelevant. |
| | 1 | Most descriptions are irrelevant, or some descriptions are completely irrelevant, e.g. transformation is unrelated to the underlying topic of the images. |
| Logical Soundness | 5 | The underlying logic of the descriptions is consistent with common sense. |
| | 4 | The overall logic is consistent with common sense, with minor flaws. |
| | 3 | There are a few obvious logical problems between the descriptions, e.g. unresonable repeating transformations. |
| | 2 | There are some obvious logical problems, e.g. the order of transformations is obviously not in line with common sense. |
| | 1 | Logic cannot be judged because of the extremely poor fluency or poor relevance leading to overall logic inconsistent with the underlying topic. |

Table 7: The VTT human evaluation guidelines.

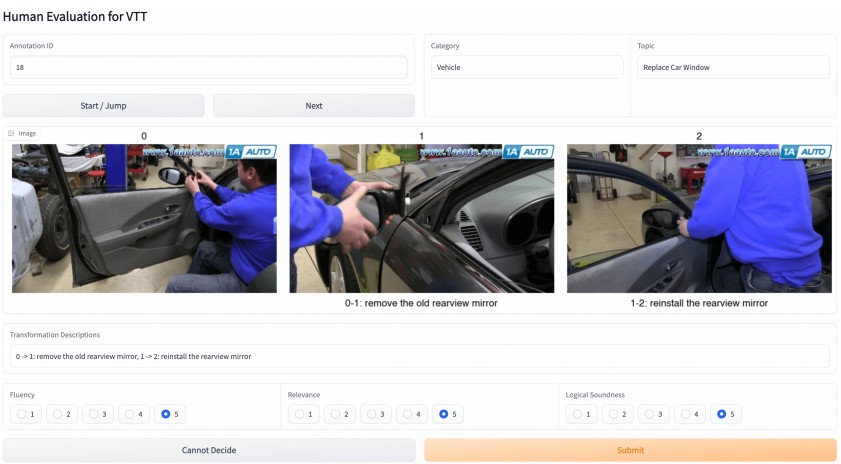

Figure 5: The web interface of human evaluation on VTT.

## B  THE CATEGORIES AND TOPICS IN VTT

Each sample in VTT has a topic and a category. All Categories and topics are shown in Table 6.

## C  EVALUATION FOR VTT

### C.1  AUTOMATIC EVALUATION

The computation of BLEU@4 follows the smooth strategy Chen & Cherry (2014) to improve the accuracy of the results. This is necessary because the descriptions in the VTT dataset are typically short, resulting in a zero score when using the original BLEU@4 method. In addition, BERT-Score is rescaled with the pre-computed baseline Zhang et al. (2020) to provide more meaningful scores with a wider range. The NLTK package [4] is used to compute BLEU@4, while CIDEr, METEOR,

---

[4] https://www.nltk.org/api/nltk.translate.bleu_score.html

ROUGE, and SPICE are computed using the code from coco-caption [5]. BERT-Score is computed using the official code [6] provided by the authors.

## C.2 HUMAN EVALUATION

Automatic evaluation metrics have limitations in reflecting the quality of the generated text, as they are uninterpretable and do not necessarily align with human evaluations van der Lee et al. (2019). To address this, we manually evaluate text quality in the VTT task using three levels of assessment. The first level assesses the fluency of the text, while the second level evaluates the relevance of each transformation description to the topic and to the images before and after. The third level assesses the logical consistency between transformation descriptions. The assessment is conducted using a 5-point Likert scale and follows the guidelines presented in Table 7. We invited 25 volunteers to evaluate major baseline models on a subset of 200 samples randomly sampled from the testing set, including one sample from each topic and two additional samples. Annotators were asked to read and follow the guidelines to assign scores. During the human evaluation process, annotators were able to view the images, the category, and the topic as references. At least two individuals evaluated each model's result for each sample. The web interface for human evaluation is shown in Figure 5 and will be included in the VTT source code.

## D TTNET

Our TTNet is inspired by human's cognitive process of transformation and existing visual storytelling models Gonzalez-Rico & Fuentes-Pineda (2018); Kim et al. (2019). In this section, we first introduce the problem formulation and the basic structure of TTNet. Then we describe how we model transformation by enhancing the model's ability to capture semantic-level differences with difference sensitive encoding, and fully utilize context to strengthen transformation reasoning with masked transformation model and auxiliary learning.

**Base structure of TTNet.** Inspired by humans and existing visual storytelling models, the first step in TTNet is independent recognition, where each image is understood independently. To achieve this, an **image encoder** $f_{\text{state}}$ is introduced to *semantize* each image into a vector, resulting in a set of state representations $V = \{v_i\}_{i=1}^{N+1} = \{f_{\text{state}}(s_i)\}_{i=1}^{N+1}$. The next step is to associate these states together to form a complete understanding of the event. To reflect this process, a **context encoder** is used. This encoder, which can be a bi-directional RNN or a transformer encoder, is denoted as $f_{\text{trans}}$ and *contextualizes* the state representations to obtain transformation representations $C = \{c_i\}_{i=1}^{N+1} = \{f_{\text{trans}}(i, V)\}_{i=1}^{N+1}$. The final step is to describe the transformations based on the existing understanding. In TTNet, this is achieved using a **transformation decoder** $f_{\text{text}}$, which can be an RNN or a transformer decoder. This decoder *textualizes* $N$ transformation representations into separate descriptions $T = \{t_i\}_{i=1}^{N} = \{f_{\text{text}}(c_{i+1})\}_{i=1}^{N}$, in an auto-regressive manner. Empirically, it was found that adding the transformation representation to the word embedding in each step is better than using it as the prefix token. The training objective is to reduce the gap between generated transformations and ground truth transformations $T^* = \{t_i^*\}_{i=1}^{N}$ by minimizing the negative log-likelihood loss, where $t_i^* = \{x_{i,l}^*\}_{l=1}^{L}$ is the ground truth description of the $i_{\text{th}}$ transformation.

$$\mathcal{L}_{\text{text}} = -\sum_{i=1}^{N} \sum_{l=1}^{L} \log p_\theta(x_{i,l}^* | x_{i,<l}^*) \tag{1}$$

Next, we introduce three strategies we used to model transformation, and we called the model that does not use these strategies as TTNet$_{\text{base}}$.

**Difference Sensitive Encoding.** To bridge the semantic gap between state differences and transformation descriptions, the first step is to enable the model to accurately identify and capture the variations between states. However, capturing differences is challenging since adjacent states often exhibit minimal variation at the pixel level. This is mainly because the scene remains almost unchanged before and after the transformation, and only certain attributes of the transformed object have changed.

---

[5]https://github.com/tylin/coco-caption
[6]https://github.com/Tiiiger/bert_score

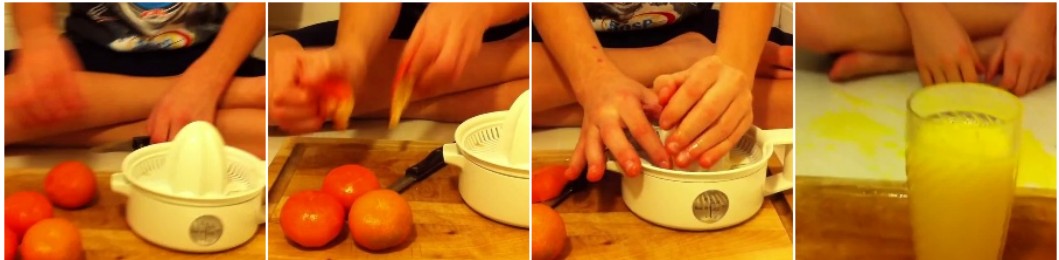

Figure 6: **The architecture of TTNet.** Images are first *semantized* into state representations in the image encoder, then *contextualized* to be transformation representations in the context encoder, and finally *textualized* into text by the transformation decoder. To better modeling transformation, difference sensitive encoding is used to capture semantic-level differences, masked transformation model and auxiliary learning are used to fully utilize context to strengthen transformation reasoning.

1. Cut both ends and remove fruit seeds.
2. Pour the egg into the bowl.
3. Pour the orange juice into the cup.

Figure 7: A failure case from TTNet$_{base}$ which has the potential to be corrected by utilizing context information.

Our intuition to solve this problem is that despite the minimal differences between states at the pixel level, there are often significant semantic differences. Therefore, we first choose CLIP Radford et al. (2021) as our image encoder to extract state representations, due to CLIP's strong semantic representation ability trained on large-scale unsupervised data. Then, we compute semantic difference features between adjacent states by subtracting the current state and the previous state representations $\Delta V = \{v_i - v_{i-1}\}_{i=1}^{N+1}$, where $v_0 = v_{N+1}$. In TTNet, we feed both state representations and the semantic difference features into the context decoder. To make the model able to distinguish these two kinds of features, we initialize two learnable types of embeddings and add them to the corresponding features.

**Masked Transformation Model.** After identifying state differences, the next challenge is to efficiently reason about the underlying transformations. For humans, one common approach is to fully utilize the context to aid reasoning rather than focusing solely on adjacent states. Therefore, we chose the transformer Vaswani et al. (2017) as the backbone of the context encoder, given its well-known ability to encode contextual information. However, in our initial experiments, we found TTNet$_{base}$ failed to fully utilize context information when reasoning about transformations. A typical example is shown in Figure 7, where TTNet$_{base}$ mistakenly identified an orange as an egg due to their similarities in the image. Nevertheless, such ambiguity can be resolved by incorporating other correct transformations. Hence, the question becomes how to enhance the model's ability to leverage contextual information. Inspired by BERT objectives, we proposed two strategies, including the masked transformation model (MTM) and auxiliary learning. Similar to the masked language model Devlin et al. (2019), the intuition behind MTM is that one transformation can be reasoned from nearby transformations. Specifically, during training, 15% of the features fed into the context encoder,

Table 8: Implementations details of baseline models and TTNet.

| Model | Image Encoder | Context Encoder | Transformation Decoder | Params |
|---|---|---|---|---|
| CST | InceptionV3 | LSTM | LSTM | 379M |
| CST* | CLIP (ViT-L/14) | LSTM | LSTM | 661M |
| GLACNet | ResNet152 | bi-LSTM | LSTM | 128M |
| GLACNet* | CLIP (ViT-L/14) | bi-LSTM | LSTM | 373M |
| DenseCap* | CLIP (ViT-L/14) | Attention | LSTM | 361M |
| TTNet$_{Base}$ | CLIP (ViT-L/14) | Transformer | Transformer | 368M |
| TTNet | CLIP (ViT-L/14) | Transformer | Transformer | 368M |

Table 9: Results of different image encoders.

| | Image Encoder | Params | Acc | B@4 | C | BS |
|---|---|---|---|---|---|---|
| ImageNet Pretrained[7] | InceptionV3 Szegedy et al. (2016) | 23M | 77.44 | 44.88 | 404.85 | 61.75 |
| | ResNet152 He et al. (2016) | 59M | 82.82 | 50.71 | 464.01 | 67.40 |
| | ViT-L Dosovitskiy et al. (2022) | 304M | 85.84 | 58.26 | 540.46 | 73.59 |
| | Swin-L Liu et al. (2021) | 196M | 86.32 | 57.36 | 531.51 | 73.03 |
| | BEiT-L Bao et al. (2022) | 306M | 87.48 | 41.57 | 370.00 | 58.80 |
| Image-text Pretrained[8] | RN50 | 39M | 73.30 | 53.35 | 491.80 | 69.79 |
| | RN101 | 57M | 75.70 | 53.78 | 495.30 | 70.08 |
| | ViT-B/32 | 88M | 76.10 | 55.21 | 510.08 | 71.27 |
| | ViT-B/16 | 86M | 80.20 | 57.73 | 534.92 | 73.37 |
| | ViT-L/14 | 304M | 83.90 | **61.22** | **570.63** | **76.25** |

including state representations and semantic difference features, are randomly masked. Empirically, we found using MTM with a 50% probability works better.

**Auxiliary Learning.** Following the target of fully utilizing context information, another strategy is focused on the global representation. BERT applied the objective of next sentence prediction (NSP) but this is not suitable for our task. However, we found humans usually try to guess the category or topic before describing transformations, e.g. cooking noodles. Therefore, we set another objective that requires TTNet to predict the category and topic from the global representation during training. Two additional cross-entropy losses $\mathcal{L}_{category}$ and $\mathcal{L}_{topic}$ can be computed from these two classification problems. The final training loss becomes a combination of $\mathcal{L}_{text}$, $\mathcal{L}_{category}$, and $\mathcal{L}_{topic}$, with adjustment factor $\alpha$ and $\beta$:

$$\mathcal{L} = \mathcal{L}_{text} + \alpha\mathcal{L}_{category} + \beta\mathcal{L}_{topic}. \tag{2}$$

# E IMPLEMENTATION DETAIL OF MODELS

## E.1 TRADITIONAL MODELS

The training process of includes standard image augmentation techniques such as random cropping and flipping, resulting in images cropped into 224×224 patches. The architectures of all baseline models are presented in Table 8.

We re-implemented CST and GLACNet based on the original papers and their released source code [9] [10]. We followed the paper for implementing the final model of DenseCap since we could not find its code. However, we used CLIP to replace DenseCap's original video encoder because it was designed for video descriptions.

---

[7]Model weights and top-1 accuracy on ImageNet of ImageNet pretrained models are from: `https://github.com/rwightman/pytorch-image-models`

[8]Pretrained weights of CLIP models are from `https://github.com/openai/CLIP` and top-1 accuracy on ImageNet is from Table 10 of the original paper.

[9]`https://github.com/dianaglzrico/neural-visual-storyteller`

[10]`https://github.com/tkim-snu/GLACNet`

```
USER:
There are {N+1} pictures of an event strip, and each picture shows one state of the event.
Write the topic of this event strip, and {N} transformations between every two adjacent panels to describe what
happened between two states that caused a state change.
Each transformation must be a phrase. Here are some examples from other pictures: "put steak on grill", "release
liquid", "add whipped cream"...

Your answer must be formatted as JSON:
{
    "topic": <the topic you wrote>,
    "transformations": [
        <the 1st transformation you wrote>,
        <the 2nd transformation you wrote>,
        ...
        <the Nth transformation you wrote>
    ]
}

ASSISTANT:
```

Figure 8: Template used to generate prompts for testing multimodal language models. The content highlighted in yellow is only used when adding a topic prediction task, it is not included in the prompt in the standard setting.

The implementation of TTNet includes a default CLIP image encoder of ViT-L/14, which is pretrained and fixed during training. We compare multiple other image encoders in Section H. The context encoder uses a transformer-based architecture consisting of two transformer encoder layers, implemented using x-transformer [11]. All transformer layers use simplified relative positional encoding Raffel et al. (2020). In the transformation decoder part, we directly borrow CLIP's tokenizer and their vocabulary list. Each transformation description is generated separately with a shared two-layer transformer decoder. The idea of adding transformation representations into word embeddings is inspired by GLACNet Kim et al. (2019) and we empirically found this way improves a lot on language influence compared with using the representation as the start token. Like the context encoder, simplified relative positional encoding is also used in the transformation decoder.

Since TTNet is greatly inspired by GLACNet, we provide a more detailed description of the relationship between these two models here. GLACNET and TTNET have a consistent overall architecture, employing an image encoder, context encoder, and decoder design. The image encoder extracts features from each image, the context encoder extracts contextual information, and finally, the decoder generates the corresponding change description. The difference lies in the implementation of different modules in GLACNET and TTNet, as seen in Table 7 of the text, from which we have extracted the relevant lines here.

We use top-$k$ top-$p$ sampling with $k = 100$ and $p = 0.9$ to generate text. The dimension of intermediate vectors, including state representations, transformation representations, and word embeddings, is set to 512. For the training loss, we set the adjustment factor $\alpha$ for $\mathcal{L}_{\text{category}}$ to 0.025 and $\beta$ for $\mathcal{L}_{\text{topic}}$ to 0.1. We use the AdamW optimizer Loshchilov & Hutter (2022), with a learning rate that warms up to 1e-4 in the first 2000 steps and then gradually decreases to 0. All models are implemented with PyTorch Paszke et al. (2019) and trained on a single Tesla A100 80G GPU card with 50 epochs. The code will be released publicly.

### E.2 MULTIMODAL LANGUAGE MODELS

To establish MLLMs performance and provide fair comparisons, we employ the exact same prompting structure as in Figure 8, in which $N$ should be replaced to the transformation number. Since

---

[11]https://github.com/lucidrains/x-transformers

Table 10: Results of different strategies of computing difference features.

| state | diff | B@4 | M | R | C | BS |
|-------|------|-----|---|---|---|-----|
| √ | - | 56.91 | 61.89 | 68.45 | 527.62 | 73.54 |
| √ | early | 60.10 | 65.16 | 70.88 | 559.78 | 75.69 |
| √ | late | **61.22** | **66.31** | **71.84** | **570.63** | **76.25** |

Table 11: Models perform worse with only adjacent states in terms of CIDEr score and re-training on them still falls short of the normal setting.

| Model | Normal | Adjacent States Only |
|-------|--------|----------------------|
| CST* | 84.90 | 49.80 |
| DenseCap* | 439.53 | 295.75 |
| GLACNet* | 508.19 | 268.49 |
| TTNet | **570.63** | 349.96 |
| TTNet (retrain) | - | **459.84** |

existing pretrained MLLMs (except Qwen) either do not support multiple image inputs or perform poorly when processing multiple images in order, we adapted the model's input requirements by collapsing the multiple images corresponding to each sample into a single one. We follow the official implementation [12] to tune LLaVA with LORA. We conduct our experiments over 50 epochs, employing a batch size of 16. The learning rate is set to 2e-5 and the warmup ratio is 0.03.

# F  MORE ANALYSES ON TTNET

## F.1  COMPARISON OF EARLY AND LATER DIFFERENCES

In the main paper, we computed the difference features in a later fusion manner, i.e., computing them on encoded image vectors to produce the semantic difference. In this section, we compare this approach with an alternative one, early fusion, which calculates pixel-level difference on raw images before feeding them to the image encoder. In TVR Hong et al. (2021), early differences were found to be more effective, while Table 10 shows the opposite result. We explain that this is because TVR involves predicting property changes on synthetic data, which relies more on pixel differences. In contrast, VTT requires event-level descriptions, placing greater emphasis on semantic distinctions.

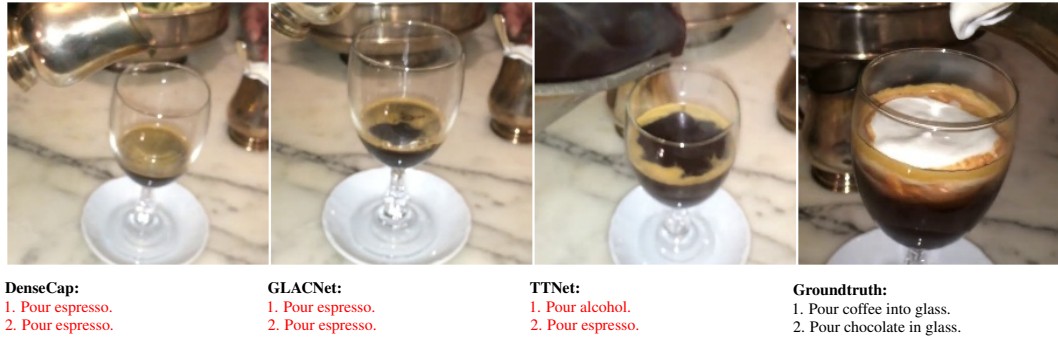

**DenseCap:**
1. Pour espresso.
2. Pour espresso.
3. Add whipped cream.

**GLACNet:**
1. Pour espresso.
2. Pour espresso.
3. Add whipped cream.

**TTNet:**
1. Pour alcohol.
2. Pour espresso.
3. Add whipped cream.

**Groundtruth:**
1. Pour coffee into glass.
2. Pour chocolate in glass.
3. Pour cream.

Figure 9: Models fail to describe unseen transformations composed by seen words.

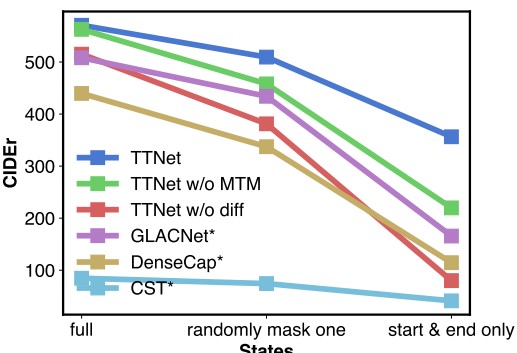

Figure 10: TTNet performs most robustly when reasoning on partial context (some states are missing).

Table 12: Models including TTNet perform worse on unseen transformation combinations.

| Model | Seen | | | | Unseen | | | |
|---|---|---|---|---|---|---|---|---|
| | C | Flu. | Rel. | Logic. | C | Flu. | Rel. | Logic. |
| CST* | 0.99 | 1.95 | 3.22 | 3.00 | 0.73 | 2.17 | 3.08 | 2.91 |
| GLACNet* | 6.21 | 4.80 | 3.90 | 3.91 | 4.11 | 4.69 | 3.70 | 3.59 |
| DenseCap* | 5.16 | 4.72 | 3.66 | 3.61 | 3.75 | 4.76 | 3.68 | 3.57 |
| TTNet$_{Base}$ | 6.02 | 4.80 | 4.08 | 4.00 | 4.40 | **4.77** | **3.99** | **3.88** |
| TTNet | **7.01** | **4.81** | **4.23** | **4.29** | **4.59** | 4.74 | 3.93 | 3.86 |

## F.2 ANALYSES ON CONTEXT MODELING

**Analyzing Context Importance for VTT.** To determine the importance of the context for VTT, we evaluated models in an independent setting where each transformation could only be reasoned from two adjacent states, without accessing other states. If context were not important, the performance of models would remain unchanged. However, Table 11 shows all four models experienced a significant performance drop. For example, TTNet's CIDEr score decreased by approximately 39%, indicating the crucial role of context in transformation reasoning. We also retrained TTNet on data constructed following the independent setting, and while performance improved, there remained a considerable gap compared to fully accessing context, further demonstrating the importance of context for VTT.

**Assessment on Utilizing Context.** Having established the importance of context, it is important to test models' ability to utilize it. We examined two settings where the provided states gradually decreased. The basic idea is that models with strong context utilization ability can compensate for missing information by relying on context. In the "randomly mask one" setting, only one state in each sample was masked, while in the "start & end only" setting, only start and end states are provided. Figure 10 demonstrates TTNet has the highest robustness as more states are missing, highlighting its exceptional ability to utilize context for transformation reasoning. Comparing TTNet to two of its variants, one without MTM and one without semantic difference features, we concluded that both MTM and semantic difference features contribute to context utilization, with the latter having a greater impact.

## F.3 ANALYSES ON TRANSFORMATION REASONING

**Assessment on Reasoning Unseen Transformation Combinations.** A robust transformation reasoning system should be able to generalize to unseen transformation combinations, where individual transformations have been seen during training, but certain combinations have not. This often occurs when there are multiple ways of achieving the same task such as cooking noodles. In VTT, more than half of the combinations in the test set are not present in the training set (532 seen vs. 559

---

[12]https://github.com/haotian-liu/LLaVA/blob/main/scripts/v1_5/finetune_lora.sh

Table 13: Results of different mask ratios used in MTM.

| mask ratio | B@4 | C | BS |
|---|---|---|---|
| 0% | 60.38 | 562.83 | 75.72 |
| 5% | 60.93 | 567.92 | 76.11 |
| 10% | 61.02 | 568.71 | 76.13 |
| 15% | **61.22** | **570.63** | 76.25 |
| 20% | 61.07 | 568.99 | 76.21 |
| 25% | 61.16 | 570.18 | **76.35** |
| 30% | 60.72 | 565.43 | 75.94 |

Table 14: Results of different sample ratios used in MTM.

| sample ratio | B@4 | C | BS |
|---|---|---|---|
| 0% | 60.38 | 562.83 | 75.72 |
| 25% | 60.39 | 562.15 | 75.63 |
| 50% | **61.22** | **570.63** | **76.25** |
| 75% | 60.96 | 567.99 | 76.00 |
| 100% | 60.95 | 568.18 | 76.10 |

unseen). To evaluate how well models can reason about unseen transformation combinations, we divided the test set into two splits: "seen" (combinations appeared in the training set) and "unseen" (new combinations). As shown in Table 12, all models perform significantly worse on the unseen combinations than on the seen ones, with TTNet's logical soundness dropping by roughly 10% (from 4.29 to 3.86), showcasing the challenge of generalization. The performance gap between TTNet, TTNet$_{Base}$, and DenseCap* on the unseen split is less significant than the gap on the seen split, implying that our strategies for modeling transformation primarily help with reasoning seen transformation combinations, while providing little benefit for reasoning unseen combinations.

**Assessment on Reasoning Unseen Language Compositions.** A robust transformation reasoning system should also be able to generalize to unseen language compositions, where individual words such as entities and actions have been seen during training, but their combinations have not. For example, successfully reasoning the unseen transformation "pour coffee" when only "pour milk" and "make coffee" appeared in the training set. According to our statistics, VTT has a high proportion of shared vocabulary, this is the major reason that VTT is designed as a natural language generation task rather than a classification task, as models have a better chance of learning common patterns from transformations with shared words. To evaluate model generalization to new language compositions, we evaluated models on several manually labeled samples from "related" tasks in CrossTask. In the example shown in Figure 9, transformations for the topic *Make Bicerin* have not appeared in VTT but are composed with seen words. However, all models failed to generate new descriptions and instead produced existing descriptions that matched the states as closely as possible. This indicates a significant limitation in the models' ability to generalize to new language compositions.

### F.4 HYPERPARAMETER TUNING OF MTM

There are two hyperparameters in the masked transformation model: the mask ratio and the sample ratio. The mask ratio is similar to that used in BERT Devlin et al. (2019), indicating the percentage of state representations and semantic difference features that are replaced with zero. After experimenting with mask ratios ranging from 0%-30%, we found 15% works best (as shown in Table 13), which is consistent with BERT's finding. The other hyperparameter is the sample ratio, which addresses the inconsistency between training and inference where no features are masked during inference. By setting the sample ratio, which is the probability that the sample will accept the masking strategy, we found a 50% probability performs best, outperforming the strategy of masking all samples used in BERT (as shown in Table 13).

USER:

Impartially assign a score for the transformation sequence ranging from 1 to 5. A transformation sequence corresponds to an event, where each transformation describes the change between two adjacent states in the event.

Each transformation in a sequence is separated by a comma.

Your scoring needs to be only considered from the perspective of logical consistency. Ignore other aspects, such as grammar, spelling, fluency, vividness, etc.

The meaning of each score is as follows:

5: The logic between the transformation descriptions is consistent with commonsense.

4: The logic between most of the descriptions is consistent with commonsense.

3: The logic between some of the descriptions is consistent with commonsense.

2: There seems to be logic between the descriptions, but it doesn't make commonsense.

1: There is no logic between the transformation descriptions, or they are completely inconsistent with commonsense.

transformation sequence: {TRANSFORMATIONS}

your score (output a numerical score directly without any extra explanation):

ASSISTANT:

Figure 11: Prompt used to evaluate logical consistency with LLM.

## G PROMPT OF LLM EVALUATION

we incorporated an automated evaluation on logical consistency using LLM. The prompt we used is shown in Figure 11.

## H COMPARISON OF DIFFERENT IMAGE ENCODERS

The quality of image encoding is crucial for subsequent reasoning and description, which determines whether the model can correctly recognize and understand the image content. Therefore, image encoder significantly impacts the overall performance of the model. In the main paper, we observe that the original version of CST and GLACNet, with Inception V3 Szegedy et al. (2016) and ResNet He et al. (2016) as image encoders, respectively, perform worse than CST* and GLACNet*. This indicates the importance of choosing an appropriate image encoder. We conduct a more detailed analysis of the image encoder by testing ten state-of-the-art image encoders, five of which were pretrained on ImageNet and five on large-scale image-text data from the CLIP variations. In the table, we report their parameter size, ImageNet top-1 accuracy, and performance on the VTT dataset. We found that when the parameter sizes were similar, models pretrained on image-text data outperformed those pretrained only on image data, e.g. ViT-L/14 vs. ViT-L. This is consistent with the existing understanding that CLIP encodes more semantic information. In addition to training data, factors that affect model performance include model size, patch size used in vision transformers, and training strategies. For example, CLIP models, which have more parameters, perform better. Although the parameter size between ViT-B/16 and ViT-B/32 is similar, ViT-B/16, which encodes finer images with smaller patch sizes, results in better image representation. BEiT-L Bao et al. (2022) has the highest accuracy on ImageNet but performs the worst among all models. We speculate that although BEiT-L has learned sufficient image pattern information, it has limitations in capturing semantic information.

## I ADDITIONAL QUALITATIVE RESULTS.

We present additional cases in Figure 12.

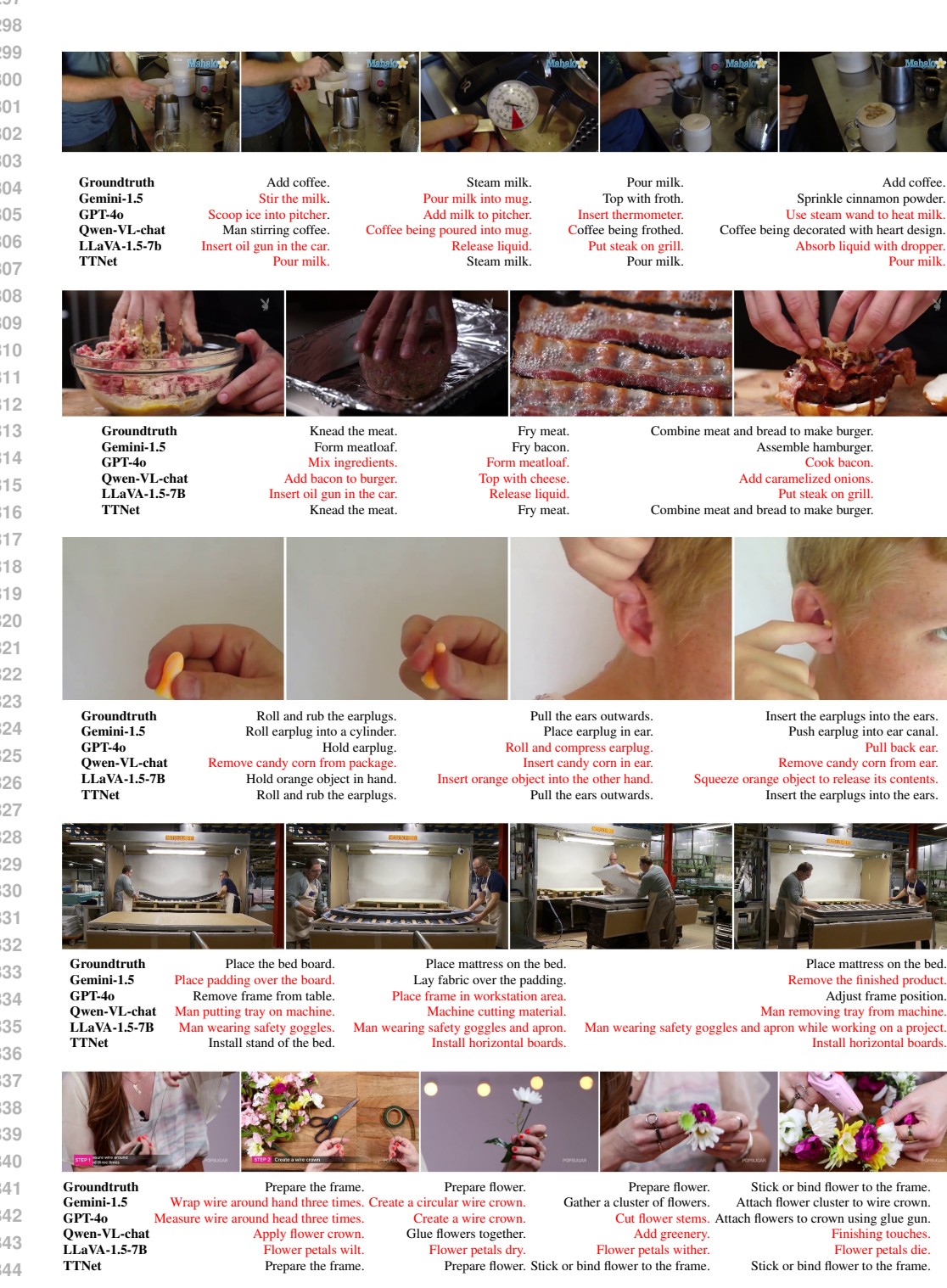

Figure 12: More Cases of MLLMs and TTNet on the VTT test data. Error outputs are marked with red.

