# OpenReview forum: "Visual Transformation Telling"
_ICLR.cc/2025/Conference — Submitted to ICLR 2025_

### Official Review · Reviewer_D7e5 · 2024-10-18

**Soundness:** 2
**Presentation:** 3
**Contribution:** 2
**Rating:** 5
**Confidence:** 3

**Summary:**

This paper introduces a task called Visual Transformation Telling (VTT), which aims to evaluate a model’s reasoning ability about the cause of state changes between two images, especially in natural language. The authors collect a large-scale dataset related to this task from instructional video datasets, conduct human verification, and create a new benchmark. This paper also tests the performance of existing open-source and commercial models and proposes a new model to address the task, which outperforms existing methods.

**Strengths:**

1. The writing of this paper is clear.

2. The experiments in this paper are comprehensive, including various quantitative tests and visual result analyses.

**Weaknesses:**

1. The difference between VTT and the existing Procedure Planning task seem not significant. Typically, Procedure Planning tasks are done on CrossTask and COIN datasets, and there is already a lot of related work on this. Moreover, both tasks are based on pre-collected temporal segment annotations from CrossTask and COIN, so the textual content should be the same. It seems like this paper simply transforms a classification problem into an open-ended text generation problem.
2. If the focus of VTT is on explanation, why not use two images instead of a reasoning chain? How does the previous reasoning process affect subsequent reasoning? When the model makes errors, are they due to the accumulation of prior mistakes or the inability to infer the current two states?
3. Specific details regarding testing models like GPT-4o need to be provided. For example, GPT-4 is a purely text-based model—how did you test it, or did you test GPT-4V? Additionally, the specific prompts and the details such as model version should be thoroughly explained.



**It is recommended to add a figure that clearly describes the differences between the task proposed in this paper and all previously related tasks.**

**Questions:**

1. How do you ensure whether a change between two images is explainable and deterministic? How do you evaluate and address the multiple possibilities of changes between two images?
2. Could more models trained on image-text interleaved data, such as InternVL-2, be evaluated?

---

### Official Review · Reviewer_9dPk · 2024-10-29

**Soundness:** 2
**Presentation:** 3
**Contribution:** 1
**Rating:** 3
**Confidence:** 4

**Summary:**

This paper proposes a new task called Visual Transformation Telling (VTT), which requires the model to generate a caption between two images in an instructional video. This paper evaluated the performance of existing models on this new task, which shows

**Strengths:**

The experiments are comprehensive. The presentation of the paper is good.

**Weaknesses:**

1. **Motivation.** The motivation to create this task is vague. It is unclear why describing the transformation between two images is more important than a single image. From the qualitative examples in the paper, it seems that only using one image is enough to generate the caption. The evidence for real-world or downstream applications is not clearly justified in the paper. Here are some questions that authors can think about when evaluating the motivation of this work: Will this new task benefit models on other tasks? What are the real-world use cases of such a task/model? Do I need a robot to understand the state transformation in **natural language format** to perform daily tasks? If the answer is only "current models cannot perform well on our task", then you need to really think about whether to do such research.

2. **Distinction with other video-language tasks.** Basically, this paper converts the existing video action recognition tasks like COIN into a video captioning task. The concept of state transformation is ambiguous, and it is hard to justify this in a better format. Furthermore, compared to the original format of COIN, the generation format of VTT poses additional difficulties in evaluation, as indicated by the bad performance of humans when using automatic metrics (Table 2).

3. **Limited Contribution.** Overall, this paper creates a task without strong motivation and claims the existing model cannot perform well on this new task. There is not much technical content in the paper, and even the data is reused from previous work. The overall contribution of this work is below the standard of ICLR.

**Questions:**

1. How did you input multiple images if the model only supports single-image input?

2. What's the performance difference between single image and multiple image inputs? For example, will models' output differ when giving $s_i$ and $s_{i+1}$ versus only using $s_{i+1}$?

3. In table 2, why Gemini gets worse performance when using multi-turn?

4. What's the fine-tuned performance of the baselines?

---

### Official Review · Reviewer_mC7F · 2024-11-03

**Soundness:** 3
**Presentation:** 1
**Contribution:** 3
**Rating:** 3
**Confidence:** 4

**Summary:**

This work creates a new benchmark which tests how well the current multimodal LLMs (MLLMs) process visual transformations (i.e. change of states) between two adjacent visual scenes, like opening/closing a door, or breaking some glass. In this benchmark, a set of open-source and closed-source models are evaluated via both human validators and automated metrics.

**Strengths:**

Significance: This work has its significance since it focuses on spatiotemporal processing, the authors discuss that some advanced MLLMs struggle to describe the given transformations (that being said, the evaluation procedure should still be revised).

**Weaknesses:**

The paper has a poor presentation.
- Fig. 1 could be improved by showing explicitly what are the inputs and the outputs in this task.
- Sec 3.1 should be linked to Fig. 1.
- The most importantly, the main text lacks details about TTNet, which makes following the paper very difficult.
- Fig. 2(d) completely could be put into the appendix (that could save some space for the TTNet).

The evaluation procedure.
- Some models achieve very high CIDEr scores. Could the authors please double check?
- While I appreciate the number of different evaulation metrics, I am skeptical about a couple of them.
- Why do we have two task-related human evaluation criterias (relevance/logical soundness) where they could be combined? If some generation is irrelevant, could the logical soundness matter for that generation? (could you please provide a counter-example if there is one?)
- Why do the authors evaluate fluency? The most of the transformation descriptions are relatively short (Fig 2.(c)) and the literature already evaluated the fluency enough previously.
- Is there a reason to avoid some evaluation procedure similar to LLM-as-judge in this work?
- It is not clear to me, how they evaluate the models. Is the evaluation per-transformation or per-scene?
- How are the images being processed by the models? For instance, as far as I know some models could process input images subsequently, are all of the tested models capable of doing the same? If not, how do the authors process input images? Do they concatenate input images?

The models.
- The number of open-source models could be increased (e.g. PaliGemma, OpenFlamingo, Idefics, InstructBLIP there are some suggestions, a newer suggestion could be Pixtral).
- Some video models could be added. Actually some of these models are pretrained using both video and image data, so they could be adapted easily (e.g. [Frozen-in-Time](https://arxiv.org/abs/2104.00650), [Video-LLaMA](https://aclanthology.org/2023.emnlp-demo.49/) and many others).

Missing some relevant literature.
- Video-Text models should be included.
- [The ViLMA benchmark's change-of-state task](https://openreview.net/forum?id=liuqDwmbQJ) is quite related to the proposed benchmark. It should be mentioned and discussed.

**Questions:**

Please see the weaknesses section.

**Details Of Ethics Concerns:**

N/A.

---

### Meta-Review · Area_Chair_7WDk · 2024-12-20

**Metareview:**

Summary: This work proposed a visual reasoning task, Visual Transformation Telling (VTT), to tell the differences between visual states transformation. A dataset with 13k samples is provided to support the task.

Strengths: The experiments are comprehensive.

Weaknesses: (1) The differences between the proposed visual transformation telling and procedure planning are not clear, challenging the motivation and novelty of the proposed task. (2) The setting of telling the differences of two images is not convincing. (3) The evaluation and results are not convincing and need further clarification.

This paper received three negative ratings as 5, 3, 3. The authors did not respond to reviewers' comments.

**Additional Comments On Reviewer Discussion:**

The authors did not respond to reviewers' comments.

---

### Decision · Program_Chairs · 2025-01-22

Reject